# The Role of Information in Enhancing Waste Sorting Capability among Consumers in Lao Cai City, Vietnam

Nguyen Thanh Phong  and Le Thi Thanh Loan *

Faculty of Economics and Rural Development, Vietnam National University of Agriculture,
Hanoi 131000, Vietnam; ntphong@vnua.edu.vn
* Correspondence: lttloan@vnua.edu.vn

**Abstract:** Waste sorting at the source has been identified as a crucial solution for sustainable waste management in developing countries like Vietnam. Numerous previous studies have investigated the factors influencing waste sorting behavior among consumers. However, there has been limited research on consumers' capability in waste sorting at source, particularly the role of information. Without sufficient consumer capability in waste sorting, contamination continues to be a widespread problem, posing a challenge to the sustainability of such programs. A survey using the ASK (Attitude-Skill-Knowledge) model was conducted with 228 participants in Lao Cai city, Vietnam, to assess how information enhances consumer capability in waste sorting. This study indicated that: (1) Both information quality and information accessibility could affect overall consumer capability toward waste sorting at the source; (2) Information accessibility is important in determining all ASK component, while interestingly, information quality plays an important role in determining consumers' attitudes toward waste sorting at the source. The study suggested some solutions for enhancing consumer capability in waste sorting program, including engaging all stakeholders, leveraging technology, educational campaigns, and regular updates on waste sorting at the source information.

**Keywords:** information quality; information accessibility; waste sorting; consumer capability; ASK model

## 1. Introduction

Since the initiation of Doi Moi policy in 1986, Vietnam's economic reforms, alongside favorable global trends, have propelled the nation from extreme poverty to a middle-income status within a single generation. Over the span of 2002 to 2022, GDP per capita surged by 3.6-fold, nearing the mark of USD 3700 [1]. Poverty rates, measured at USD 3.65 per day in 2017 PPP, dwindled from 14 percent in 2010 to a mere 3.8 percent by 2020 [1]. As the country experienced economic growth, waste management has emerged as a significant issue due to the rapid and unprecedented increase in waste production. Projections indicate that waste generation will triple over the next 15 years, posing a substantial challenge. At present, Vietnam lacks the necessary infrastructure to manage this surge effectively: 70% of waste is disposed of in landfills with limited adherence to environmental standards, while the remainder is either incinerated or improperly discarded in natural environments, contributing to a significant portion ultimately finding its way into the sea [2]. It is not surprising that managing municipal solid waste in Vietnam presents a complex challenge, and implementing waste sorting at the source could be a commendable solution to enhance effectiveness and reduce costs. Globally, municipal solid waste source-separated collection is recognized as a leading strategy for waste management, with many cities adopting it for sustainable development [3–13].

Vietnam initiated its municipal solid waste sorting at source program in Hanoi in June 2001, followed by pilot programs in Hanoi, Danang, and Ho Chi Minh City over the past decades. Recently, several Vietnamese cities like Hoi An, Danang, and Lao Cai

have embraced the waste sorting at source program. However, these initiatives warrant reconsideration for potential enhancements and broader implementation. The 14th National Assembly of Vietnam recently enacted the 2020 Law on Environmental Protection to fortify ongoing environmental preservation endeavors and introduce fresh perspectives and innovations in waste management. Coming into effect on 1 January 2022, this legislation underscores the significance of source sorting at the heart of waste collection and the trajectory of the circular economy. The circular economy, hailed as an intelligent environmental protection strategy, harmonizes seamlessly with source sorting. By closing the loop, waste resources are reimagined as inputs for production, necessitating strategic realignment and reengineering of existing infrastructure, technologies, and logistics to facilitate circularity. Source sorting streamlines the reuse process by sorting waste, facilitating its reuse. The Vietnamese government anticipates that this approach will foster greater adoption of municipal solid waste separation at source and source reduction, as failing to do so may result in exorbitant waste treatment costs.

There are numerous studies addressing the promotion of waste sorting behavior among consumers, originating from both Vietnam [14–24] and around the world [25–30]. These studies emphasize the influence of psychological factors [16,24,31,32], economic incentives [24,28,33], as well as infrastructure and government support [16,24,34–36]. In addition to these factors, [37] underscores the significance of consumers capability in the waste sorting at source program. The literature suggests that without adequate consumer capability in waste sorting, although sorting may occur, contamination remains a prevalent issue.

Lao Cai city, located in the northwest region of Vietnam, faces the challenge of managing approximately 200 tons of solid waste daily due to rapid urban development [38]. Since December 2015, the city has implemented the municipal solid waste sorting at source program, guiding consumers to separate waste into organic, inorganic, and recyclable categories. Consumers receive information and guidance on proper waste disposal and sorting practices. Collected waste is treated accordingly: organic waste is composted, while inorganic waste is buried in designated landfills. The program aims to significantly reduce daily waste generation, with hopes of addressing the growing waste issue in the city. Success relies heavily on widespread social support for effective implementation. According to the Report of the Urban Environment Company of Lao Cai Province, in 2021, the rate of waste classification at the source in the city exceeded 90%; however, the rate of contaminants in the waste separated remains quite high, which hinders the process of composting organic waste at the city's waste treatment plant. Therefore, enhancing consumers' capability in waste sorting is important to ensure the success of waste sorting at the source program.

The popular capability framework, known as the ASK (Attitude-Skill-Knowledge) model, was originally proposed by Benjamin Bloom and his colleagues in 1956. According to this model, consumer capability consists of three main components: attitude, skill, and knowledge. Attitude pertains to consumers' emotions and feelings, encompassing their perspectives on natural and social phenomena. Skills, often referred to as manual or physical skills, involve the execution of tasks and the application of existing knowledge. Knowledge refers to cognitive abilities, understood as what individuals accumulate or acquire, serving as a fundamental capacity necessary for task completion.

Although the ASK model has found extensive application in various research fields such as education, marketing, human resource management, and pesticide reduction, there is limited literature exploring its application in waste sorting. Only one study has applied the ASK model in waste sorting, focusing only demographic characteristic to determine consumer capability and neglecting other important external factors such as the role of information [37]. Therefore, the recommendations did not effectively enhance consumers' sorting capability by strengthening the role of information. Numerous studies have emphasized the importance of information in promoting environmental behavior [39–43], as well as recycling behavior at the source [41–43]. This study aims to

fill an academic gap by meticulously examining the role of information in promoting consumer capability through the ASK model in waste sorting at the source, thereby proposing appropriate solutions to enhance consumer capability in waste sorting at the source in the future. The research was conducted in Lao Cai city, Vietnam.

## 2. Theoretical Framework

### 2.1. Definition of Consumers' Capability toward Waste Sorting at the Source

The ASK model, initially introduced by Benjamin Bloom and his colleagues in 1956 [44], is a widely recognized capability framework. This model suggests that consumer capability comprises three primary elements: attitude, skill, and knowledge. Attitude encompasses consumers' emotions and perspectives on natural and social phenomena. Skills, sometimes labeled as manual or physical abilities, involve task execution and the application of acquired knowledge. Knowledge, on the other hand, refers to cognitive abilities, representing what individuals accumulate or learn, and it is essential for task completion. According to [37], the ASK model has been applied in various research areas, including education, human resource development, pesticide reduction, etc.

In the ASK model, Knowledge and Attitude components are quite commonly investigated when seeking to understand consumer recycling or their waste sorting behavior. In term of knowledge, studies have strongly confirmed that consumer knowledge is crucial for promoting recycling and waste sorting because it empowers individuals to make informed decisions about their waste disposal [25,41,45–50]. Understanding the principles and benefits of recycling, including which materials can be recycled, how to prepare them for recycling, and how to deposit recyclable items, is key. When they are aware of what can be recycled/sorted, as well as understanding the benefits of recycled/ sorted activities, their actions are significantly impacted [49]. By promoting the necessary knowledge, recycling or waste sorting initiatives can craft targeted messages and address misconceptions to remove these barriers. Some literature has pointed out that clear and consistent information about what materials belong in which bin is essential. Confusing labelling or unclear sorting guidelines can lead to contamination of recyclables, reducing their overall value. Without clear knowledge of what is recyclable, consumers might mix trash with recyclables, negating the program's environmental benefits [25,51]. Moreover, consumer knowledge about the benefits of waste sorting at source program is also very crucial to support the consumers to separate waste in the correct way. For example, they know that organic waste could produce compost fertilizer, so therefore sorting organic waste correctly can be good behavior they perform. Therefore, consumer knowledge campaigns can ensure everyone understands the local sorting system. Educated consumers are more likely to embrace recycling activities. Informative campaigns can significantly boost participation rates in recycling programs [16,52]. In term of attitude, consumer attitudes toward waste sorting behavior encompasses the belief, feelings, and evaluations individuals have about the practice of sorting waste at the point of generation [16,34,53,54]. These attitudes significantly influence their willingness to participate in waste sorting activities, comply with program guidelines, and contribute to overall waste management efforts [16]. Consequently, understanding consumer attitudes is crucial for determining their capability in effective waste sorting. Positive attitudes towards waste sorting translate into a stronger willingness to participate. If consumers believe it is important and beneficial, they are more likely to put in the effort to sort their waste correctly. In addition, sorting waste can require some additional effort, especially compared to simply throwing everything in one bin. A positive attitude helps people overcome this inconvenience and see waste sorting as a worthwhile action. Waste sorting at source is most effective when it becomes a habit. Positive attitudes create a sense of responsibility and environmental consciousness, making waste sorting a natural part of consumers' routines. Many studies have pointed out that consumers with a strong attitude are more likely to sort waste diligently. By fostering positive consumer attitudes, waste sorting initiatives can encourage long-term participation and make a significant impact on waste management [16,24,25,48]. In term of skill, studies revealed that consumer skills

are a factor within the variable "perceive behavioral control", in other words, the ability to control behavior [9,22,55]. Accordingly, the skills in waste sorting or recycling are the ability to accurately sort waste in an easy, time-saving, and cost-effective way (proficiency, less time to do it, instruct others on how to sort, less cost). Other studies mentioned that utilizing sorting techniques also improved proficiency by employing appropriate methods to separate different types of waste efficiently, either at the source or during disposal. Skills to minimize waste generation improve through practices such as composting organic waste, reusing items, or opting for products with minimal packaging. In addition, included within the skills of consumers toward waste sorting should be an awareness of safety protocols for handling and disposing of hazardous waste materials, including proper storage and disposal methods to prevent environmental contamination and health risks. Skills are also important in the case of community engagement. Consumers have the ability to participate in community initiatives, such as neighborhood clean-up events or waste reduction campaigns, to promote collective responsibility and awareness of waste management issues. The highest-level skills can show the willingness of consumers to stay informed about advancements in waste sorting technologies, evolving recycling practices, and sustainable lifestyle choices to continually improve waste sorting skills and reduce environmental impact [16,50,56].

### 2.2. Information and Consumers' Capability toward Waste Sorting at the Source

Various studies have highlighted the role of information in promoting environmental behavior [39–43]. Previous research has shown that adequate knowledge of what to recycle, provided through recycling information and feedback, significantly enhances recycling behavior at the source [41–43]. Some studies have indicated that publicity and promotion positively influence household recycling behavior and observed a positive association between recycling knowledge and household recycling behavior [41]. All these studies suggest that information about waste recycling, whether in the form of feedback, publicity, promotion, or a well-designed communication strategy, is an effective tool for engaging and enhancing recycling behavior at the source [41–43].

With the importance of information in promoting waste recycling or waste sorting behavior at the source, information quality and information accessibility are crucial. Improving the quality and accessibility of recycling information is essential for fostering better recycling behaviors. Detailed, easily accessible, and well-structured information helps in overcoming barriers and motivates individuals to participate more actively in recycling programs.

Studies have emphasized the importance of information quality [41]. Accordingly, information quality relies on information that is detailed, accurate, and relevant to the recycling context. High-quality information can help consumers understand what can be recycled, the benefits of recycling, and the proper methods for sorting waste. High-quality information also highlighted the importance of specific, well-structured information; breaking down recycling information into categories like "what, when, and where" can make it more comprehensive and actionable, therefore increasing recycling behavior.

Information accessibility is also noted to be crucial. Previous studies indicated that the ease with which individuals can access recycling information plays a significant role in their willingness to participate in recycling programs [42,43]. Information needs to be readily available through various channels such as social media, educational institutions, and community programs. Research involving college students found that information from media and educational sources positively affected their recycling attitudes and behaviors, while interpersonal communication was less effective [43]. Providing both written and visual information can cater to different preferences and enhance understanding. An experimental study showed that both forms of information could positively change recycling behaviors, although visual information sometimes had a more substantial impact on specific actions like purchasing behavior [42].

## 3. Methodology

### 3.1. Study Site

The waste separation at source program was formally introduced in Lao Cai city, Lao Cai province, Vietnam in December 2015, marking the beginning of a series of events leading to its expansion beyond the city to other areas of Lao Cai province. The initial stage of implementation included Sapa city and Bat Xat district on 15 March 2016 and 15 January 2017, respectively. After the pilot phase in 2021, the waste sorting at source program expanded to other districts in the province. This study focuses solely on the implementation of the program in Lao Cai city, Vietnam. The city covers 282.13 square kilometers and includes 10 wards and 7 communes. As of 2019, Lao Cai city had a population of 130,671 people, with a population density of 463 individuals per square kilometer. The research for this study was conducted in six wards and communes of the city, including Kim Tan and Coc Leu wards, which were part of the initial pilot phase of the waste sorting at source program in December 2015. Bac Cuong and Binh Minh wards implemented the waste sorting at source program in March 2016, while Thong Nhat and Xuan Tang communes implemented it in February 2017.

### 3.2. Survey Administration

The face-to-face survey, conducted in Lao Cai city, Vietnam, from January to February 2021, employed a two-stage sampling method. Initially, a pre-test survey was conducted with 10 randomly selected consumers to assess the questionnaire's clarity and effectiveness. Based on the findings, minor adjustments were made to ensure the questions were better understood by respondents. The second stage involved stratified random selection of households in the city, resulting in an effective sample size of 228 due to COVID-19 restrictions. One adult consumer from each household represented the sample. Participants were informed about the research, gave consent, and were assured of anonymity and data confidentiality.

### 3.3. The Questionnaire

The construction of the three-section questionnaire utilized a foundation of diverse literature exploring recycling proficiency, sorting skills, and other facets of the ASK model across different research domains [16,21,23,24,41–43,45,47,49]. The initial section comprised inquiries aimed at gathering consumers' socio-demographic data. Following this, the second section encompassed questions pertaining to attitudes, skills, and knowledge. Lastly, the final section focused on gathering insights into consumers' perceptions regarding information dissemination concerning the waste sorting at source program in Lao Cai city (Appendix A).

#### Attitude

Consumer attitude toward the waste sorting at source program refers to the belief, feelings, and evaluations individuals hold regarding the practice of separating waste at the point of generation [14,16,24,28,34,45,49,51,57,58]. Consumer attitudes influence their willingness to engage in waste sorting activities, adhere to program guidelines, and contribute to waste management efforts. Therefore, this component is crucial in determining consumers' capability. In this study, the consumer attitude component is measured by eight questions on Likert 5-level scales (ranging from Strongly Disagree to Strongly Agree). These questions focus on households' preferences and concerns regarding the waste sorting at the source program.

#### Skills

A study examined individuals' skills in consumer waste recycling in Italy [59]. According to the study, skills are a component of the variable "perceived behavioral control", which essentially refers to the capability to control behavior. Therefore, skills in waste sorting primarily involve the ability to accurately sort waste easily, without consuming much time or cost (proficiency level, time efficiency, instructing others in sorting, cost

efficiency) [57,59]. From this explanation, the questions of this study in determining skills toward waste sorting at the source behavior focus on assessing the proficiency of households in waste sorting, such as their ability to accurately sort waste, time and cost efficiency in sorting, and coordination/instruction of family members or neighbors in waste sorting. The question set is designed with 5 Likert-scale questions ranging from 1 (Strongly Disagree) to 5 (Strongly Agree).

*Knowledge*

When consumers have less knowledge about their task than they think they have, they might inadvertently separate their waste incorrectly. The consumer knowledge questionnaire is divided into two parts: (1) Recognition of organic waste/inorganic waste/recycle waste, consisting of 17 questions corresponding to knowledge about 17 common types of waste (include various vegetables, fruits, leftovers, tea leaves, eggshells, plastic items, cans, rubber items, leather items, glassware, paper, various flowers, food wrappers, various fabrics, small branches and leaves, paper packaging, and cardboard boxes); these questions are designed with answer options of True/False [41,60–62]; (2) Understanding the benefits of waste sorting [16], comprising six Likert-scale questions with five levels, ranging from 1 (Strongly Disagree) to 5 (Strongly Agree).

*Information quality*

The quality of information is accessed through various statement questions [41,42], including: (i) Information published about waste sorting at the source program is reliable and accurate; (ii) Information published about the waste sorting at the source program is easy to understand; and (iii) Information published on waste sorting at source program is comprehensive and complete.

*Information accessibility*

The accessibility of information is accessed through various statement questions [41–43]; (i) It is easy to receive information about the consequences of not sorting waste at the source; (ii) It is easy to obtain information about instructions on how to classify and recycle waste in daily life; (iii) It is easy to access information about the city's penalties for not sorting waste at the source; and (iv) It is easy to access information about the benefits of sorting waste at the source.

*3.4. Consumer Capability Calculation*

The data collected is considered to assess the consumer's capability in waste sorting at the source. First, for questions in True/False/Don't know format, True is scored as 1 point, False/Don't know is scored as 0 points. Likert scale questions were evaluated from 1 to 5 corresponding to the five levels of the scale. Thus, Attitude assessment scores range from 1 to 40, Skill assessment scores range from 1 to 25, and Knowledge assessment scores range from 1 to 47. Finally, the converted scores entered into the quantitative economic model were calculated as percentages by taking the achievable scores of A, S, and K group for consumers, dividing by the maximum possible score, and multiplying by 100. Previous studies had also employed this calculation method and incorporated post-calculation information into quantitative economic models [63].

The consumers' capability toward waste sorting at the source program will be estimated as an important result of this study. Refs. [37,49] has calculated ASK but investigation the determinants was investigated separately for each component of ASK. Therefore, we cannot see how the factors influencing in overall of consumer's capability. Moreover, refs. [37,49] did not investigate the latent factors influencing capability; therefore, the recommendation for policy makers in promoting consumers' capability toward waste sorting at the source will be limited.

### 3.5. Exploratory Factor Analysis

Exploratory Factor Analysis (EFA) is a statistical technique used to identify the underlying structure of a set of variables. It helps in understanding the relationships between variables and uncovering latent factors that may be influencing the observed variables. Through conceptualization, this study has indicated that information quality and information accessibility are crucial in enhancing consumers' capability toward waste sorting at the source. Before conducting EFA, the study is needed to ensure that the data are suitable for factor analysis. We checked the adequacy of sample size, the correlation matrix among variables, and the suitability of the data for factor analysis using the Kaiser–Meyer–Olkin (KMO) test and Bartlett's test of sphericity. Next, we choose an extraction method to identify the underlying factors. Since common methods include Principal Component Analysis (PCA), we used PCA with varimax rotation. After that, we used Kaiser's criterion (eigenvalue > 1) to determine the number of factors to retain. After running the EFA, we examined the factor loadings, which represent the correlation between each variable and identified factors. Variables with high loadings on a particular factor are considered to be strongly associated with that factor. Based on the pattern of factor loadings, we interpreted and labeled each factor according to the variables that load most strongly on it. This step involves understanding the underlying constructs represented by the factors. After that, we evaluated the reliability of the factors using measures like Cronbach's alpha for internal consistency. Additionally, we assessed the convergent and discriminant validity of the factors to ensure that they are distinct from each other and measured the intended constructs. The two variables explored during the EFA step are continually investigated as independent variables in the next econometric model. This step will allow the study to explore how the latent factors identified through EFA predict outcomes of interest.

### 3.6. Multiple Regression Analysis

We investigated four models where the dependent variables are the capability score, attitude score, skill score, and knowledge score towards waste sorting at the source, which appear as continuous variables; therefore, a multiple regression model was used.

The regression model used can be explicitly specified as:

$$Y = \beta_0 + \beta_i X_i + \varepsilon_i$$

where Y is the consumer capability score/attitude score/skill score/knowledge score toward waste separation at the source, $X_i$ is independent variable (factors which retain from EFA), $\beta_0$ is the intercept, $\beta_i$ is the coefficients of the independent variable $X_i$, and $\varepsilon_i$ is the random error. The software SPSS version 22.0 was used to perform the EFA and regression analysis.

## 4. Results and Discussion

### 4.1. The Consumers' Socio-Economic Backgrounds

Table 1 presents the socio-economic backgrounds of the consumers. Among those who participated in the interview, 42.54% were female and 57.46% were male. The consumers' ages ranged from 15 to 70 years old, with the middle age group (21–50 years old) representing 48.68%. Notably, 46.05% of the sampled consumers have lived in Lao Cai city for over 21 years, suggesting they have have a comprehensive understanding of their community. Additionally, 77.63% of the respondents reported living in households of 1 to 4 people, with an average household size of 4.13, indicating the prevalence of nuclear families in Vietnamese cities. Furthermore, 55.7% of the sample had children in their households. Most consumers had received more than 12 years of education, comprising 46.49% of the total survey sample.

**Table 1.** The socio-economic characteristics of consumers in Lao Cai city (*n* = 228).

| Criteria | Categories | Frequency | Percent |
|---|---|---|---|
| Gender | Female | 97 | 42.54 |
| | Male | 131 | 57.46 |
| Age group | Young (15–20) | 89 | 39.04 |
| | Middle (21–50) | 111 | 48.68 |
| | Old ($\geq$51) | 28 | 12.28 |
| Number of years living time in the city | $\leq$15 years | 27 | 11.84 |
| | 16–20 years | 96 | 42.11 |
| | $\geq$21 years | 105 | 46.05 |
| Consumers' household size | 1–4 people | 177 | 77.63 |
| | 5–10 people | 51 | 22.37 |
| Presence of children in consumers' households | Yes | 127 | 55.7 |
| | No | 101 | 44.3 |
| Education level | $\leq$5 years | 8 | 3.51 |
| | 7 years | 55 | 24.12 |
| | 9 years | 38 | 16.67 |
| | 10 years | 21 | 9.21 |
| | 12 years | 106 | 46.49 |

*4.2. Reliability Coeficient and Exploratory Factor Analysis Results*

Cronbach's alpha was employed to measure the internal consistency of the items in a group. Previous literature suggests that 0.7 is an acceptable reliability coeficient [64]. In this study, the realiability coefficients for the information quality is 0.861 while the realiability coefficients for the information accessibility is 0.833 (Table 2). Therefore, we can confirm that both coeficients are acceptable realiability.

**Table 2.** Factor analysis for aspect and criteria.

| Aspect | Criteria | Factor Loading | Cronbach Alpha |
|---|---|---|---|
| Information quality | Information published about waste sorting at the source program is reliable and accurate | 0.895 | 0.861 |
| | Information published about the waste sorting at the source program is easy to understand | 0.881 | |
| | Information published on waste sorting at source program is comprehensive and complete | 0.758 | |
| Information accessibility | It is easy to receive information about the consequences of not sorting waste at the source | 0.851 | 0.833 |
| | It is easy to obtain information about instructions on how to classify and recycle waste in daily life | 0.876 | |
| | It is easy to access information about the city's penalties for not sorting waste at the source | 0.715 | |
| | It is easy to access information about the benefits of sorting waste at the source. | 0.673 | |

The results of the Bartlett's test were significant ($p$ = 0.000) and the KMO value was 0.831, higher than 0.5, indicating that the value was appropriate to be used for the EFA process. The factor loadings were higher than 0.75, indicating the sigificance of the criteria (Table 2). Eigenvalue is 1.158, greater than 1, which strongly confirmed that the two factors, namely information quality and information accessibility, are retained [64].

### 4.3. Consumers' Capability toward Waste Sorting at the Source

Among the 288 consumers surveyed, the capability score, also known as the average ASK score, was 72.30%, with scores ranging from 52.36% to 89.70%. The mean attitude score was 74.10%, with scores ranging from 47.5% to 100%. The mean skill score was 68.98%, with scores ranging from 32% to 96%, and the mean knowledge score was 73.84%, with scores ranging from 53.19% to 91.49% (Table 3).

**Table 3.** Summary of dependent variables.

| Dependent Variables | Mean | Std.Dev | Min | Max |
|---|---|---|---|---|
| Capability score | 72.30 | 7.62 | 52.36 | 89.70 |
| Attitude score | 74.10 | 10.68 | 47.5 | 100 |
| Skill score | 68.98 | 12.41 | 32 | 96 |
| Knowledge score | 73.84 | 6.25 | 53.19 | 91.49 |

### 4.4. Econometric Results

Table 4 presents the results of ecometric estimation from the four models.

**Table 4.** Econometric estimation of the four models.

| Independent Variables | Capability Score | | | Attitude Score | | | Skill Score | | | Knowledge Score | | |
|---|---|---|---|---|---|---|---|---|---|---|---|---|
| | B | S.E. | Sig. | B | S.E. | Sig. | B | S.E. | Sig. | B | S.E. | Sig. |
| Information quality | 1.051 | 0.478 | 0.029 ** | 1.5244 | 0.694 | 0.029 ** | 0.984 | 0.786 | 0.212 | 0.645 | 0.401 | 0.109 |
| Information accessibility | 2.356 | 0.478 | 0.000 *** | 1.799 | 0.694 | 0.010 ** | 3.724 | 0.786 | 0.000 *** | 1.545 | 0.401 | 0.000 *** |
| Constant | 72.308 | 0.477 | 0.000 *** | 74.100 | 0.693 | 0.000 *** | 68.982 | 0.784 | 0.000 *** | 73.842 | 0.400 | 0.000 *** |
| Model summary | Prob > F = 0.0000 R-squared = 0.1146 Adj R-squared = 0.1067 | | | Prob > F = 0.0036 R-squared = 0.0487 Adj R-squared = 0.0403 | | | Prob > F = 0.0000 R-squared = 0.0963 Adj R-squared = 0.0883 | | | Prob > F = 0.0002 R-squared = 0.0718 Adj R-squared = 0.06350 | | |

Note: *** Significant $p$_value at 1% level and ** at 5% level.

Overall, all four models are statistically significant. The good fit of these models are demonstrated by the R-squared values, which indicate that 11.46%, 4.87%, 9.63%, and 7.18% of the variance in the dependent variable are explained by information quality and information accessibility factors in the Average ASK score model, Attitude score model, Skill score model, and Knowledge score model, respectively.

In term of coefficients, the results from Table 4 show that both information quality and information accessibility are useful predictors, with a significant positive relationship with the average ASK score. Information accessibility significantly affects the average ASK score at the 1% level ($p$-value = 0.000 < 0.01), while information quality significantly predicts the average ASK score at the 5% level ($p$-value = 0.029 < 0.05). Information accessibility has a strongly positive effect on attitude score, skill score, and knowledge score, with significant values of 5% ($p$-value = 0.010 < 0.05), 1% ($p$-value = 0.000 < 0.01), and 1% ($p$-value = 0.000 < 0.01), respectively. Meanwhile, information quality plays an important role in predicting the attitude score, with significance at the 5% level ($p$-value = 0.029 < 0.05), but it is not significant in predicting the skill score and knowledge score.

## 5. Discussion

### 5.1. Role of Information Accessibility in Enhancing Consumer Capability toward Waste Sorting at Source

The findings from the study indicated the importance of information accessibility in predicting consumer capability toward waste sorting at the source. In this study, the accessibility of information is gauged through various statements, including: (i) ease of receiving information about the consequences of not sorting waste at the source; (ii) ease of obtaining instructions on waste classification and recycling in daily life; (iii) ease of accessing information about the city's penalties for not sorting waste at the source; and (iv) ease of accessing information about the benefits of waste sorting at the source. It indicates the importance of ease of receiving information related to waste sorting behavior.

The literature underscores the significance of information regarding regulatory effectiveness in promoting pro-environmental behavior [16,24]. According to the literature, consumers expressed frustration with their efforts being in vain: they sorted their waste, only to observe waste collectors mixing it with unsorted waste or waste from non-compliant neighbors. Consequently, they are less inclined to participate in the program. Therefore, accessing information about punitive measures or regulations for non-compliance with the waste sorting program is crucial [16,24]. Accordingly, it is necessary to improve the involvement of all relevant stakeholders [65], such as socio-political organizations, including women's unions, youth unions, and others, to disseminate information about the waste sorting at the source program. Currently, Vietnam is beginning to apply digital transformation in providing information and monitoring waste sorting behaviors at the source [52]. This solution could promote easier information accessibility, thereby enhancing consumer capability regarding waste sorting at the source in the city.

Previous research has demonstrated the importance of recognizing types of waste [17,49,51] as well as the benefit of waste recycling and waste sorting at the source [14,45,51,58,66]. Therefore, accessibility to information related to types of waste and the benefits of the waste sorting at source program is important for enhancing consumer capacity regarding the program. In Vietnam, guideline information for waste sorting appears in leaflets, posters, and other materials provided by the local authorities [52]. However, there is no research yet that emphasizes which kinds of information sources should be distributed to consumers to maximize the effectiveness of information accessibility in enhancing consumer capability.

### 5.2. Role of Information Quality in Enhancing Consumer Capability toward Waste Sorting at the Source

According to the results in Table 4, information quality can affect both attitude scores and capability scores. Therefore, enhancing information quality could improve consumer attitudes and, as a result, increase consumer capability regarding waste sorting at source. In this study, the quality of information is evaluated through several statements, such as: (i) The information provided about the waste sorting at the source program is reliable and precise; (ii) The information provided about the waste sorting at the source program is easy to comprehend; and (iii) The information provided about the waste sorting at the source program is thorough and complete. To enhance information quality, it is necessary to ensure that the information provided for waste sorting is reliable, precise, comprehensible, thorough, and complete.

Previous research has indicated the importance of information quality by highlighting the need for detailed, accurate, and relevant content in the recycling context. Research indicated that enhancing information quality is crucial for boosting recycling participation. This entails clearly specifying the what, when, and where of recycling, as well as providing guidance on how to recycle effectively. Moreover, when crafting and disseminating recycling information, it is important to educate recipients about the significance and rationale behind recycling. Continuous updates to recycling information are necessary to accommodate shifts in waste generation behaviors and seasonal trends, considering the influence

of timing and context on recycling habits. To encourage and support recycling efforts, mechanisms for offering feedback on recycling performance are essential. Additionally, efforts to provide recycling information and communication should prioritize accessibility to recycling programs, catering to both current recyclers and those yet to adopt recycling practices [41]. Research conducted in Hoi An city, Vietnam, also suggested eliciting frequent feedback from consumers to strengthen positive attitudes. Additionally, providing encouraging reminders to individuals who hold negative views towards the waste sorting at the source program was recommended [16].

## 6. Recommendations and Future Research

The study highlights the importance of information accessibility and quality. To enhance the importance of information accessibility, the study suggests some recommendations for local authorities in Lao Cai city to enhance consumer capability in the waste sorting at the source program: (1) Engage all stakeholders: improve the involvement of all relevant stakeholders, such as socio-political organizations; (2) Leverage technology: increase the involvement of technology companies to vigorously promote digital transformation in providing information and monitoring waste sorting behaviors at the source. To enhance the quality of information, the recommendations include: (3) Educational campaigns: implement educational campaigns and materials that highlight the environmental benefits of waste sorting; (4) Regular updates: establish systems for regularly updating waste sorting at the source information to ensure its accuracy and relevance; and (5) Providing encouraging reminders: encourage individuals who hold negative views towards waste sorting.

Further study should explore what kinds of information sources are most effective in being distributed to consumers to enhance their capability regarding information accessibility. Future research should also focus on the crucial role of information in every waste sorting program to ensure the sustainability of these programs, especially in developing nations.

**Author Contributions:** Conceptualization, N.T.P. and L.T.T.L.; Methodology, L.T.T.L.; Investigation, N.T.P.; Writing—review & editing, L.T.T.L.; Project administration, L.T.T.L. All authors have read and agreed to the published version of the manuscript.

**Funding:** This research is funded by Vietnam National Foundation for Science and Technology Development (NAFOSTED) under grant number 502.01-2020.318.

**Institutional Review Board Statement:** Not applicable.

**Informed Consent Statement:** Informed consent was obtained from all subjects involved in the study.

**Data Availability Statement:** Data are contained within the article.

**Acknowledgments:** The authors would like to thank the Vietnam National Foundation for Science and Technology Development (NAFOSTED) for their financial support under grant number 502.01-2020.318. We would also like to thank Andres Fiona Rose Norada and Raquel M. Balanay for their encouragement and their kind help in proofreading the English language of the manuscript.

**Conflicts of Interest:** The authors declare no conflicts of interest.

## Appendix A. Questionnnaire

I.    ASK component

    *A.*    Attitude (5 levels Likert scale questions from Strongly Disagree to Strongly Agree)

       1.   Waste sorting is good for me
       2.   Waste sorting is good for everyone
       3.   Waste sorting is good for environment
       4.   Waste sorting is good task to perform
       5.   Waste sorting is interesting

6. Waste sorting should be expanded to all cities in the countries
7. I have good attitude towards waste sorting
8. I need to sort waste

*S.* Skill (5 levels Likert scale questions from Strongly Disagree to Strongly Agree)

1. Accurately sort waste
2. Time efficiency in sorting waste
3. Cost efficiency in sorting waste
4. Coordination/instruction of family members in sorting waste
5. Coordination/instruction of neighbors in sorting waste

**K. Knowledge**

Recognize type of organic/inorganic and recycle waste (True/False questions)

| | | | | | |
|---|---|---|---|---|---|
| 1. | Vegetables | 2. | Cans | 3. | Food wrappers |
| 4. | Fruits | 5. | Rubber items | 6. | Fabrics |
| 7. | Food leftovers | 8. | Leather items | 9. | Branches and leaves |
| 10. | Tea leaves | 11. | Glassware | 12. | Paper packaging |
| 13. | Eggshells | 14. | Paper | 15. | Cardboard boxed |
| 16. | Plastic items | 17. | Flowers | | |

Recognize the benefit of waste sorting at the source (5 levels Likert scale questions from Strongly Disagree to Strongly Agree)

1. Practicing waste sorting would improve your image
2. Practicing waste sorting would create a small income
3. Practicing waste sorting could reduce waste generated
4. Waste sorting could produce compost fertilizer
5. Waste sorting could save budget for waste collection and treatment of the city
6. Waste sorting could reduce the pressure for the city's landfill

II. **Information factors**

*2.1.* **Information quality** (5 levels Likert scale questions from Strongly Disagree to Strongly Agree)

1. Information published about waste sorting at the source program is reliable and accurate
2. Information published about the waste sorting at the source program is easy to understand
3. Information published on waste sorting at the source program is comprehensive and complete.

*2.2.* **Information accessibility** (5 levels Likert scale questions from Strongly Disagree to Strongly Agree)

1. It is easy to receive information about the consequences of not sorting waste at the source
2. It is easy to obtain information about instructions on how to classify and recycle waste in daily life
3. It is easy to access information about the city's penalties for not sorting waste at the source
4. It is easy to access information about the benefits of sorting waste at the source.

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
