# Peer review of "The Role of Information in Enhancing Waste Sorting Capability among Consumers in Lao Cai City, Vietnam"

_sustainability, doi:10.3390/su16146244_

Round 1

Reviewer 1 Report

Comments and Suggestions for Authors

1. The title should be revised, e.g. at the Source.

2. The macro results are listed in the abstract and conclusion sections. It is suggested that some numerical results should be supplied in the abstract and conclusion sections.

3. There are no figures in the manuscript.

4. The references on the utilization and sorting of waste are insufficient, and some references should be cited and analyzed, e.g., DOI: 10.1016/j.fuel.2023.129925; 10.1016/j.fuel.2022.124388.

5. There are some short paragraphs (even two lines) in the manuscript and should be merged.

6. There are some syntax errors, sentence errors and other English Language errors that must be corrected. Possibly, a native English Language speaking Scientists should be employed for the final editing.

Comments on the Quality of English Language

Moderate editing of English language required.

Author Response

  1. The title should be revised, e.g. at the Source. 

    We have rewritten our paper title as follows.

    The Role of Information in Enhancing Waste Sorting Capability among Consumers in Lao Cai City, Vietnam

    Line 1-2
  2. The macro results are listed in the abstract and conclusion sections. It is suggested that some numerical results should be supplied in the abstract and conclusion sections. 

    The abstract includes numerical results.

    Line 15-18
  3. There are no figures in the manuscript. Yes, we describe our data in four tables, which clearly show the mean, standard deviation, minimum, and maximum values.
  4. The references on the utilization and sorting of waste are insufficient, and some references should be cited and analyzed, e.g., DOI: 10.1016/j.fuel.2023.129925; 10.1016/j.fuel.2022.124388. 

    Thank you for your suggestion; we have added the recommended reference (10.1016/j.fuel.2022.124388) and other references.

    Line 61-62 and Reference number 14 to 18 in the reference list.
  5. There are some short paragraphs (even two lines) in the manuscript and should be merged. 

    We have combined the short paragraphs.

    Line 114 to Line 115
  6. There are some syntax errors, sentence errors and other English Language errors that must be corrected. Possibly, a native English Language speaking Scientists should be employed for the final editing. The revised version has been proofread by a native English speaker in the field of Economics.

Reviewer 2 Report

Comments and Suggestions for Authors

in table one, please add another raw presenting the male. as per your interpretation the female fraction was 57% , accordingly the 43%f also should be considered.

numbers with decimal points should have the English decimal point (.) not (,)

Table 4 need to be formatted specially the left column.

the author mentioned that issuing effective laws and detailed regulations for enforcement considered as a part of information about measured to enhance waste separation. actually, this is not correct, regulation is another tool which is completely different than information and attitude of residents.

the author recommendations are not mainly based on the findings of the study i suggest linking the recommendations to the study findings.

the authors need to provide the questionnaire as a separate file.

the author, mentioned that the survey conducted in two stages, what was the results of the first stages and type of improvement resulted from this stage.

did the authors conduct the questionnaire in person or distributed it over the internet.

there are many typos and formatting issues in the entire manuscript, English needs to be improved. 

Author Response

1. In table one, please add another raw presenting the male. as per your interpretation the female fraction was 57% , accordingly the 43%f also should be considered. 

Thank you very much for your suggestion. We have improved Table 1 by more clearly indicating the socio-economic characteristics of consumers in Lao Cai city.

Line 339 to Line 349

2. Numbers with decimal points should have the English decimal point (.) not (,). We have revised the entire paper.

3. Table 4 need to be formatted specially the left column. We have revised Table 4.

4. The author mentioned that issuing effective laws and detailed regulations for enforcement considered as a part of information about measured to enhance waste separation. actually, this is not correct, regulation is another tool which is completely different than information and attitude of residents. 

Thank you for your comment. Based on your feedback, we have revised our paper to improve the clarity of our explanation regarding the role of information related to effective laws and detailed regulations.

Line 396 to Line 402.

5. The author recommendations are not mainly based on the findings of the study i suggest linking the recommendations to the study findings. 

Thank you for your comment. Based on your suggestion, we have revised our recommendation by indicating more clearly solutions related to the discussion section.

Line 453-454 and Line 458-459

6. The authors need to provide the questionnaire as a separate file. The authors have provided a questionnaire in the Appendix.

7. The author, mentioned that the survey conducted in two stages, what was the results of the first stages and type of improvement resulted from this stage. 

Thank you for your comment. We have added the results of the first stage of sampling.

Line 223-225

8. Did the authors conduct the questionnaire in person or distributed it over the internet. 

Thank you for your comment. From January to February 2021, despite Covid-19 restrictions, we were still allowed to conduct a face-to-face survey. However, the restrictions limited the sample size for our research. We have added this information to the manuscript.

Line 221-226

9. There are many typos and formatting issues in the entire manuscript, English needs to be improved. We have made efforts to correct typos and formating issues. The revised version has been proofread by a native English speaker in the field of Economics.

Reviewer 3 Report

Comments and Suggestions for Authors

The manuscript titled "The Role of Information in Enhancing Waste Sorting Capability at the Source among Consumers in Lao Cai City, Vietnam." After reviewing it, we have identified several areas for improvement to enhance clarity, rigor, and overall impact. The abstract should be more concise, clearly stating the research problem, methodology, key findings, and implications. The introduction needs a more detailed explanation of the ASK model and the literature gap your study addresses. The literature review should include more recent studies and diverse perspectives on factors influencing waste sorting behavior, as well as advancements in waste management technologies. The methodology section requires detailed descriptions of data collection and analysis methods, including the sampling method, survey instrument, validation process, and statistical analysis techniques. Results should be presented clearly with tables, figures, and statistical significance values, summarized with structured subheadings for different ASK model components. In the discussion, interpret results in the context of existing literature, highlighting practical implications and providing specific recommendations for policymakers and practitioners. The conclusion should succinctly summarize main findings, implications, and suggest future research areas. Ensure all references are consistently cited and formatted per the journal's guidelines. Address grammatical errors, awkward phrasing, and typographical errors throughout the manuscript, ensuring consistent terminology. Review statistical analysis for accuracy, supporting all claims with appropriate evidence, and ensure all graphs and tables are clear, correctly labeled, and adequately referenced. Specific data points needing correction include the GDP per capita figure for 2022 on page 2, line 30, which requires a citation, and clarification of the exact amount of waste managed daily in Lao Cai city on page 2, line 69, ensuring consistency throughout the manuscript. 

Comments on the Quality of English Language

Address grammatical errors, awkward phrasing, and typographical errors throughout the manuscript, ensuring consistent terminology

Author Response

1. The abstract should clearly state the research problem, methods, key findings, and implications. Currently, it is a bit verbose. 

Thank you for your suggestion, the revison is made.

Line 12-14

2. The introduction provides a good overview but could benefit from a more detailed explaination of the ASK model and its relevance to waste sorting. Need to provide a brief overview of how the ASK model has been used in similar contexts and also clarify the gap in the current literature that this study aims to fill. 

We have added a sentence to emphasize the literature gap.

Line 96-97

3. While the literature review touches on relevant studies, it can be more comprehensive by including more recent studies and diverse perspectives. Expand on the various factors infuencing waste sorting behavior (e.g., psychological, economic, infrastructural) and discuss recent advancements in waste management technologies and their impacts on consumer behavior. 

Thank you very much for your comment. we have added a technological aspect to the discussion of this study, highlighting the importance of digital transformation in disseminating information and monitoring waste sorting.

Line 408-409

4. Methodology section should provide a detailed explanation of how data was collected and analyzed. Specify the sampling method used to select participants. Describe the survey instrument and its validation process and explain the statistical methods used to analyze the data. 

Thank you very much for your comment. We have added an explanation of the survey method, indicating that respondents were selected through stratified random sampling. Line 226-line 227

Face to face survey has been indicated.

Line 222

Statistical software to analyze data has been indicated in Line 338-339.

5. Ensure that the results are presented clearly, with appropriate use of tables, figures, and statistical analysis. Summarize key findings in a more structured format (e.g., subheadings for different components of the ASK model). Include statistical significance values where applicable. 

We have indicated Table 2 and Table 3 to present the results.

Line 357, line 362 and line 370.

6. The discussion should interpret the results in the context of the existing literature and highlight the practical implications of the findings. Discuss how the findings align or contrast with previous studies. Provide specific recommendations for policymakers and practitioners based on the results. 

We have tried to improve our discussion and have referenced the relevant literature.

Line 399 to 405.

7. The conclusion should succinctly summarize the main findings and their implications, and suggest areas for future research. Reiterate the importance of information quality and accessibility in enhancing waste sorting capability. Suggest practical steps for improving waste sorting programs based on the study’s findings. Identify any limitations of the study and propose directions for future research.

The conclusion has been revised from lines 451 to 468.

8. Check for grammatical errors, awkward phrasing, and typographical errors throughout the manuscript.

We have made efforts to correct typos and formating issues throughout the manuscript.

9. Ensure that terminology is used consistently (e.g., waste separation vs. waste sorting). Review the statistical analysis for accuracy. Ensure that all claims are supported by apporpriate statistical evidence.

If stating that information quality significantly affects consumer attitudes, provide the corresponding p-value or confidence interval.

We have revised the entire manucript and have used the terminology “sorting” instead of “separating”.

We have indicated p-value in the results’ explanation from Lines 379 to 388.

10. On page 2, line 30, the GDP per capita figure for 2022 needs a citation. 

The reference has been cited at Line 29.

11. On page 2, line 69, clarify the amount of waste managed daily in Lao Cai city and ensure this figure is consistent throughout the manuscript.  

The manuscript has indicated the information of waste generated daily life in Lao Cai city (approximately 200 tons). Line 69

Round 2

Reviewer 1 Report

Comments and Suggestions for Authors

1. The References 15 and 16 are the same.

2. The format of the references should be consistent.

3. In the fourth comment, the related references are wrong, the correct ones should be cited as follows,

Cheng, M.N. Zhang, Y.H. Zhang, B. Lin, H.J. Zhan, and H.J. Zhang, “A novel renewable collector from waste fried oil and its application in coal combustion residuals decarbonization,” Fuel, vol. 323, no. 1, 2022, p. 124388, 2022, doi: https://doi.org/10.1016/j.fuel.2022.124388.

G. Cheng, M.N. Zhang, Y. Lu, H.J. Zhang, and E.V. Lau, “New insights for improving low-rank coal flotation performance via emulsified waste fried oil collector,”Fuel, vol.357, p. 129925, 2024, doi: https://doi.org/10.1016/j.fuel.2023.129925.

Comments on the Quality of English Language

Minor editing of English language required.

Author Response

  1. The References 15 and 16 are the same.  (We have mergered the 2 references.)
  2. The format of the references should be consistent. (We have revised)
  3. In the fourth comment, the related referencesare wrong, the correct ones should be cited as follows, (We have revised)

Cheng, M.N. Zhang, Y.H. Zhang, B. Lin, H.J. Zhan, and H.J. Zhang, “A novel renewable collector from waste fried oil and its application in coal combustion residuals decarbonization,” Fuel, vol. 323, no. 1, 2022, p. 124388, 2022, doi: https://doi.org/10.1016/j.fuel.2022.124388.

  1. Cheng, M.N. Zhang, Y. Lu, H.J. Zhang, and E.V. Lau, “New insights for improving low-rank coal flotation performance via emulsified waste fried oil collector,”Fuel, vol.357, p. 129925, 2024, doi: https://doi.org/10.1016/j.fuel.2023.129925.

Round 3

Reviewer 1 Report

Comments and Suggestions for Authors

The third and fourth references are wrong and should be revised as follows,

[3] G. Cheng, M.N. Zhang, Y.H. Zhang, B. Lin, H.J. Zhan, and H.J. Zhang, “A novel renewable collector from waste fried oil and its application in coal combustion residuals decarbonization,” Fuel, vol. 323, no. 1, 2022, p. 124388, 2022, doi: https://doi.org/10.1016/j.fuel.2022.124388.

[4] G. Cheng, M.N. Zhang, Y. Lu, H.J. Zhang, and E.V. Lau, “New insights for improving low-rank coal flotation performance via emulsified waste fried oil collector,”Fuel, vol.357, p. 129925, 2024, doi: https://doi.org/10.1016/j.fuel.2023.129925.

Comments on the Quality of English Language

Minor editing of English language required.

Author Response

Thank you for your comments. Our revision has been improved.

The third and fourth references are wrong and should be revised as follows,

[3] G. Cheng, M.N. Zhang, Y.H. Zhang, B. Lin, H.J. Zhan, and H.J. Zhang, “A novel renewable collector from waste fried oil and its application in coal combustion residuals decarbonization,” Fuel, vol. 323, no. 1, 2022, p. 124388, 2022, doi: https://doi.org/10.1016/j.fuel.2022.124388.

[4] G. Cheng, M.N. Zhang, Y. Lu, H.J. Zhang, and E.V. Lau, “New insights for improving low-rank coal flotation performance via emulsified waste fried oil collector,”Fuel, vol.357, p. 129925, 2024, doi: https://doi.org/10.1016/j.fuel.2023.129925.